# An Experimental Study on the Impact of Layer Height and Annealing Parameters on the Tensile Strength and Dimensional Accuracy of FDM 3D Printed Parts

**DOI:** 10.3390/ma16134574

**Published:** 2023-06-25

**Authors:** Jelena R. Stojković, Rajko Turudija, Nikola Vitković, Filip Górski, Ancuţa Păcurar, Alin Pleşa, Alexandru Ianoşi-Andreeva-Dimitrova, Răzvan Păcurar

**Affiliations:** 1Faculty of Mechanical Engineering, University of Niš, Aleksandra Medvedeva 14, 18000 Niš, Serbia; rajko.turudija@masfak.ni.ac.rs (R.T.); nikola.vitkovic@masfak.ni.ac.rs (N.V.); 2Faculty of Mechanical Engineering, Poznan University of Technology, Piotrowo 3 STR, 61-138 Poznan, Poland; filip.gorski@put.poznan.pl; 3Department of Manufacturing Engineering, Faculty of Industrial Engineering, Robotics and Production Management, Technical University of Cluj-Napoca, Blv. Muncii, No. 103-105, 400641 Cluj-Napoca, Romania; ancuta.costea@tcm.utcluj.ro; 4Department of Mechatronics and Machine Dynamics, Faculty of Automotive, Mechatronics and Mechanical Engineering, Technical University of Cluj-Napoca, Blv. Muncii, No. 103-105, 400641 Cluj-Napoca, Romania; alin.plesa@mdm.utcluj.ro (A.P.); alexandru.ianosi@mdm.utcluj.ro (A.I.-A.-D.)

**Keywords:** PLA, PETG, PETGCF, annealing, layer thickness, regression model, mechanical testing

## Abstract

This study investigates the impact of annealing time, temperature, and layer height on the tensile strength and dimensional change of three 3D printing materials (PLA, PETG, and carbon fiber-reinforced PETG). Samples with varying layer heights (0.1 mm, 0.2 mm, and 0.3 mm) were annealed at temperatures ranging from 60–100 °C for 30, 60, and 90 min. Tensile tests were conducted, and regression models were developed to analyze the effects of these parameters on tensile strength. The models exhibited high accuracy, with a maximum deviation of only 5% from measured validation values. The models showed that layer height has a significantly bigger influence on tensile strength than annealing time and temperature. Optimal combinations of parameters were identified for each material, with PLA performing best at 0.1 mm/60 min/90 °C and PETG and PETGCF achieving optimal tensile strength at 0.1 mm/90 min/60 °C. PETGCF demonstrated smallest dimensional change after annealing and had the best modulus of elasticity of all the materials. The study employed experimental testing and regression models to assess the results across multiple materials under consistent conditions, contributing valuable insights to the ongoing discussion on the influence of annealing in 3D-printed parts.

## 1. Introduction

The process of additive manufacturing (AM), also known as 3D printing, involves building a three-dimensional object by adding layers of material until the desired shape is achieved. This technique utilizes various technologies, materials, and machines to produce products or prototypes [1]. One popular and widely used AM technique is Fused Deposition Modeling (FDM), which involves extruding layers of melted thermoplastic material through a nozzle to build up the final shape layer by layer. While FDM can produce complex geometries and is cost-effective and easy to use, the mechanical properties of FDM-printed parts may not always be as strong or durable as those made through traditional manufacturing processes with metals or other materials. Factors that affect FDM part properties include layer orientation, bonding, infill density, material type, temperature, layer thickness, print speed, and nozzle diameter. However, researchers are continuously investigating ways to enhance FDM part properties such as mechanical strength, accuracy, and surface roughness, by utilizing different post-treatments, printing parameter optimization techniques, material characterization, material reinforcement, etc. Studies on mechanical properties have examined the tensile strength, elasticity, and other mechanical properties of FDM-printed parts [2]. Material characterization studies have focused on the thermal and mechanical properties of different FDM materials such as PLA, ABS, nylon, PETG, and others [3,4,5,6,7,8]. Surface roughness studies have aimed to identify the factors that affect the surface roughness of FDM-printed parts and develop methods to improve it.

Post-treatment research has investigated techniques to improve the quality and functionality of FDM-printed parts [9]. Optimization studies have investigated the effects of printing parameters on mechanical properties, surface quality, and overall performance [10,11,12,13] as well as the relationships between printing parameters and final properties. Other studies have explored the optimization of printing parameters to improve the mechanical properties and surface quality of FDM-printed parts [14,15,16,17]. Material reinforcement studies have focused on adding fibers, fillers, particles, or other materials to the base FDM material to improve mechanical properties and functionality [18,19,20,21,22]. Additionally, FDM printing has been applied in various fields, such as tissue engineering in biomedical engineering, where scaffold architectures have been created using FDM printing [23,24,25,26,27,28,29]. Overall, FDM printing research is multidisciplinary and covers many fields. It continues to provide valuable insights into the capabilities and limitations of this additive manufacturing technique.

Based on the literature review, the thickness of FDM layers has a significant impact on various aspects of printing, including mechanical properties, dimensional accuracy [30,31,32], print time, and surface quality [33,34]. A lower layer thickness is beneficial for achieving better dimensional accuracy and improving the surface finish quality of printed parts, while a higher layer thickness is recommended for reducing print time. When it comes to mechanical properties, the tensile strength is a frequently examined property and is mostly influenced by layer thickness and orientation during fabrication. To achieve optimal tensile strength, numerous studies suggested using lower values of layer thickness [35,36,37,38,39,40]. Thus, the layer thickness is a crucial parameter that needs to be carefully optimized to achieve specific print quality goals, which is why it was incorporated it in the study. In addition, annealing has been extensively studied as a post-processing method to enhance the mechanical properties of FDM printed parts [41,42]. While it can improve certain properties, it may also cause warping or distortion if it is not performed correctly.

Therefore, material, annealing, and printing conditions should be considered before deciding to use annealing as a treatment. The effectiveness of annealing mainly depends on the annealing time and temperature [43], and different materials may respond differently to the process. Most studies have focused on PLA materials, but other materials such as ABS, PETG, PLACF, PETGCF, and PEEK, among others, have also been explored [44]. These studies typically vary the annealing temperature and time as primary variables to investigate their effects on specific mechanical properties. In Table 1 the overview of the most relevant scientific papers that deal with annealing process and its effects on mechanical properties of 3D-printed parts is presented.

The existing literature on FDM 3D printing has provided valuable insights into the impact of various process parameters on the mechanical properties and dimensional accuracy of printed parts. However, despite the progress made, there are still gaps in the understanding of the role that controlled annealing processes and layer height play in achieving optimal results. To address this issue, we incorporated varying annealing times, temperatures (including temperatures just above the glass transition temperatures of different materials), and layer heights for 3D prints. We maintained consistent experimental conditions by utilizing identical 3D printer, standardized setups, the same annealing conditions, controlled cool-down times, and other critical factors. Determining the annealing parameters involved analyzing experiences from researchers in relevant papers, conducting a comprehensive literature review, considering material manufacturer recommendations, and drawing upon our own previous experiences. Temperature selections included the crystallization temperatures of each material, with variations above and below these values. Furthermore, we explored lower and higher temperatures to assess their impact on material characteristics and validate observed temperature trends. The selection of annealing time and other parameters followed the same principle.

By conducting experiments with three input parameters, including layer height, annealing time, and annealing temperature, as well as three different materials (PLA, PETG, and PETGCF), this study intends to address these gaps in the literature and provide a more comprehensive understanding of the factors influencing the mechanical properties and dimensional accuracy of FDM 3D printed parts. PLA material was chosen because it is one of the most commonly used materials for FDM printing. PETG and PETGCF were selected as materials due to their excellent characteristics and wide range of applications for which there is insufficient data on the effects of annealing in the existing literature. Additionally, it was valuable to compare how the reinforcement with carbon fibers affects the mechanical properties and dimensional stability of these two materials under both pre-annealing and post-annealing conditions.

Upon thorough examination of the literature, it becomes evident that only a limited number of studies have considered the combined influence of layer height (LH), annealing time (TTA), and annealing temperature (TA) in their experimental setups. Furthermore, the range of values explored for each parameter has been relatively restricted. In contrast, some studies utilized parameter values that were not relevant or appropriate. Therefore, based on these observations, we determined an optimal range of parameter values to investigate. The majority of research studies use 100% infill and different types of infill patterns. A pattern has been chosen that, based on the literature and previous made testing made by the authors, would be suitable for functional parts. For the infill percentage, we decided on 75% as it falls within the recommended range (50–100%) for 3D-printed models expected to possess strength and withstand higher loads in real-world conditions. When considering the practical implications of 3D printing, it has been observed that using a 100% infill percentage does not significantly enhance strength compared to lower infill percentages. However, it significantly increases printing time and cost. Taking this into consideration, it was decided to examine the behavior of printed parts with a 75% infill percentage. This infill percentage is widely applicable for various use cases, particularly for functional components manufactured using FDM 3D printing. The authors’ choice was based on both the findings from existing literature and our own previous experiences. The aim of this study is to contribute to the growing body of research on the effects of annealing on FDM printed parts and offer valuable insights into the behavior of these materials under the same annealing conditions. This assessment aims to contribute to understanding of how these materials perform and their viability for practical use.

## 2. Materials and Methods

The selected materials for the study included polylactic acid (PLA), polyethylene terephthalate glycol (PETG), and carbon fiber reinforced PETG (PETGCF). PLA, a thermoplastic monomer, is produced from renewable resources such as corn starch or sugar cane, unlike most plastics which are made using fossil fuels. PETG, thermoplastic polyester, boasts notable chemical resistance, durability, and formability for manufacturing. PETG was developed from PET by incorporating glycol at the molecular level to enhance various chemical properties. Although PETG contains the same monomers as PET, it is stronger, more durable, more impact-resistant, and can withstand higher temperatures. PLA and PETG are among the most commonly used filament materials in 3D printing, and each has its own set of advantages and disadvantages. PLA has low melting and glass transition temperatures and is not influenced by its surroundings during the printing process, making it easy to print using any FDM printer. However, its primary drawback is its lack of strength, resulting in brittle 3D-printed models due to limited layer-to-layer adhesion. PETG, on the other hand, is very strong, durable, and resistant to high temperatures, water, and chemical solvents, making it a suitable choice for parts exposed to harsh conditions or high physical stress. Incorporating carbon fiber (CF) into PETG should theoretically improve PETGCF’s material properties compared to PETG. For this study, PrimaSelect PLA PRO (Prima Creator, Malmö, Malmö, Sweden) [62], with a glass transition temperature of 60 °C and melting point (140–160 °C), Devil Design PETG (Devil Design, Mikołów, Poland) [63] with a glass transition temperature of 80 °C and melting point (220–250 °C) and Black Element PETG with 30% carbon fiber (Black Element, Shenzhen, China) and melting point (220–250 °C) [64] with a glass transition temperature of about 78 °C have been considered.

The materials were printed on a Wanhao Duplicator i3 Plus 3D printer (Wanhao, Jinhua, China) at three different layer heights, 0.1 mm, 0.2 mm, and 0.3 mm. For printing PETGCF, a steel nozzle was used due to its unique properties. To anneal the specimens, an industrial-grade oven was used at various temperatures ranging from 60 °C to 100 °C for different periods, namely 30 min, 60 min, and 90 min. To establish a baseline for the tensile strengths, additional specimens were printed for each layer height and material, without undergoing any annealing process. Table 2 illustrates all the alternatives considered in the experiment.

The specimen used for the tensile testing was created in accordance with the ASTM D638-14 standard [65]. Type 1 geometry was selected as the preferred type of specimen for this standard, and its geometry is depicted in Figure 1.

The Wanhao Duplicator i3 Plus 3D printer was used for printing as it is capable of printing all the selected materials. This printer has a build volume of 200 × 200 × 180 mm, a maximum nozzle temperature of 260 °C, and a print bed temperature of 100 °C. It uses a filament size of 1.75 mm and has a layer resolution of 0.1 mm. The printer specifications are outlined in Table 3, and Figure 2a shows the printer. Printing was carried out in batches of five specimens, as shown in Figure 2b. The UltiMaker Cura 5.2.2. software (Ultimaker, Utrecht, The Netherlands) was used for slicing, as illustrated in Figure 2b, and printing parameters for the materials are listed in Table 4.

The purpose of the study was to assess the capability of the selected materials for creating functional and durable 3D printed parts under realistic conditions. To emulate such conditions, specimens were printed with a 75% infill percentage. The use of a cubic infill pattern was justified, as it is the most employed pattern for producing strong functional parts that utilize infill patterns such as cubic subdivision, octet, and quartet cubic.

To anneal the printed specimens, an industrial-grade oven was utilized, with the thermometer carefully positioned inside to ensure the required temperature was reached. To ensure an even and consistent temperature throughout the heating area, specimens were only placed in the oven after allowing them to reach the required temperature for half an hour. To prevent the specimens from sticking to the steel plate during the annealing process, a thin layer of sand was placed between the samples and the plate. A visual representation of the oven, along with the thermostat and specimens placed inside, is provided in Figure 3.

To perform the tensile testing, the Shimadzu Table-top AGS-X 10 kN universal testing machine (Shimadzu, Kyoto, Japan) was utilized. This machine comprises various components, including a rigid frame, a movable crosshead, test fixtures that serve as jaws to hold the specimen, force sensors, and a control unit, among others. Figure 4a illustrates the universal testing machine, and Figure 4b illustrates some of the specimens to be tested. Specifications for the Shimadzu Table-top AGS-X 10 kN universal testing machine are detailed in Table 5.

To measure change in nominal dimensions, calipers were utilized. However, due to the inherent limitations of 3D printing, which involves high temperatures and inconsistent printed dimensions, absolute dimensions were not considered (absolute dimensions refer to measurements taken after annealing). Instead, relative dimensions were used and expressed as percentages to define change from nominal dimensions after annealing. The relative changes were calculated for the thickness of the specimen (*T*), which is 3.2 mm, the width of the narrow section of the specimen (*Wc*), which is 13 mm, and the overall length of the specimen (*LO*), which is 165 mm, as illustrated in Figure 1. The following equations were used to calculate the relative change percentage:(1)$$\Delta Wc=100-\frac{Wc\;before\;annealing}{Wc\;after\;annealing}\times 100\%$$
(2)$$\Delta T=100-\frac{T\;before\;annealing}{T\;after\;annealing}\times 100\%$$
(3)$$\Delta LO=100-\frac{LO\;before\;annealing}{LO\;after\;annealing}\times 100\%$$

To determine and calculate this change in the dimensional accuracy of the specimens after annealing, measurements were taken at two different time points. The first measurement was taken after printing and cooling the specimens for an hour (*Wc* before annealing, *T* before annealing, *LO* before annealing), while the second measurement was taken an hour after the annealing process to allow the specimens to fully cool (*Wc* after annealing, *T* after annealing, *LO* after annealing). Measuring instruments used included a digital caliper for the thickness and width of the narrow section and a caliper with nonius for the overall length. The specimens’ overall length of 165 mm exceeded the maximum length capacity of the digital caliper, which was 150 mm. Therefore, a caliper with nonius was utilized to measure the overall length.

In this study, regression models were employed to investigate the impact of various input parameters on the tensile strength of 3D-printed PLA, PETG, and PETGCF parts. For all the materials, empirical models in the forms of second-order polynomials with interactions were developed, through use of Excel software, based on the measurements obtained through previously explained method of tensile testing. The models’ adjusted R-squared values (R^2^_adj_) were 0.7979, 0.8305, and 0.7789 for PLA, PETG, and PETGCF, respectively. To verify the reliability of the regression models, additional tests with various specimens were performed, each with different layer heights, annealing times, and temperatures. The predicted tensile strengths based on the models were compared to the actual measured values obtained from the confirmation experiments. It was found that the regression models were highly accurate, with a maximum deviation of only 5% from the measured values, with the majority of predictions being under 3% off the measured values. These results demonstrate that the developed regression models are dependable and can be utilized to evaluate the influence of input factors on tensile strength, offering crucial insights into enhancing mechanical properties of 3D printed parts.

## 3. Results and Discussion

The mechanical strength of 3D printed parts is determined by various factors such as layer height, annealing time, and annealing temperature. This study investigates the effects of these variables on the tensile strength and dimensional change of 3D printed parts made from three different materials: PLA, PETG, and PETGCF, using regression models. To determine the effects of layer height (LH), annealing time (TTA), and annealing temperature (TA) on tensile strength, separate regression models were developed for each material. The mathematical regression models for PLA, PETG, and PETGCF are presented below.
PLA: Tensile Strength = 30.70314286 − 1.348680612 × LH + 0.186613358 × TTA + 0.373598425 × TA + 0.127909091 × LH^2^ − 0.251181818 × TTA^2^ − 0.270487013 × TA^2^ − 0.031022727 × LH × TTA − 0.043891742 × LH × TA + 0.000590472 × TTA × TA(4)
PETG: Tensile Strength = 30.46869841 − 1.605755651 × LH − 0.131014688 × TTA − 0.483246622 × TA + 0.768636364 × LH^2^ + 0.039090909 × TTA^2^ − 0.134577922 × TA^2^ + 0.047386364 × LH × TTA + 0.406638292 × LH × TA − 0.060424954 × TTA × TA(5)
PETGCF: Tensile Strength = 23.83793651 − 2.451571064 × LH + 0.506168091 × TTA − 0.365493993 × TA + 0.573409091 × LH^2^− 0.038863636 × TTA^2^+ 0.871428571 × TA^2^ − 0.335795455 × LH × TTA + 0.720375677 × LH × TA + 0.312162793 × TTA × TA(6)

The coefficients for layer height (LH), annealing time (TA), and annealing temperature (TTA) provide valuable insights into their influence on tensile strength for different materials. Among all materials tested, LH has a significantly greater impact on tensile strength than TTA and TA, as indicated by the much larger coefficient for LH. This trend is particularly evident in PETGCF. Furthermore, all materials exhibit negative coefficients for LH, indicating that an increase in LH results in a decrease in tensile strength and vice versa. This can be observed in Figure 5, where the tensile strengths of PLA material for different layer heights are presented, showing a decreasing tendency.

The TTA coefficient for PLA is slightly positive (0.186613358), indicating that a slight increase in the annealing time will lead to a slight increase in the tensile strength. Similarly, the TTA coefficients of PETG and PETGCF are −0.131014688 and 0.506168091, respectively. Comparing these coefficients shows that changing the annealing time will have more influence on PETGCF material than PLA and PETG.

When observing the TA coefficients of PLA and PETG (0.373598425 and −0.483246622, respectively), it is evident that annealing temperature has more influence on tensile strength than annealing time (TTA), which suggest that changing the annealing temperature will have a greater effect on tensile strength than changing annealing time.

This is not the case for PETGCF material, as its coefficient for TA is −0.365493993, which is less than the coefficient of TTA (0.506168091). This means that for PETGCF, annealing time has a greater influence on tensile strength than annealing temperature.

Regression models are useful for predicting output parameters based on input parameters, but they do not provide the complete picture. Figure 6 provides a more detailed illustration of how the annealing temperature affects the given materials. For all materials, only one or two annealing temperatures yielded a relatively significant increase in tensile strength. In Figure 6a, tensile strength increases at temperatures of 80 °C or 90 °C for PLA material (and depending on the LH sometimes at 100 °C). For example, for LH, TTA, and TA of 0.1 mm, 60 min and 90 °C, respectively, tensile strength increased by 1.22 N/mm^2^ compared to the base tensile strength (with no annealing conducted, and LH of 0.1 mm). For PETG, the increase in tensile strength can be seen at temperatures around 60 or 70 °C (Figure 6b). For example, for LH, TTA, and TA of 0.1 mm, 60 min and 70 °C, respectively, tensile strength increased by 0.91 N/mm^2^ compared to the base tensile strength (with no annealing conducted, and LH of 0.1 mm). However, when it comes to PETGCF (Figure 6c), it is more challenging to detect a clear tendency. For 0.2 mm layer height, it can be noted that the increase in tensile strength of 0.9 N/mm^2^, 1.38 N/mm^2^, and 0.06 N/mm^2^ was at 70 °C, for TTA of 30 min, 60 min, and 90 min, respectively, compared to the base tensile strength. For 0.3 mm layer height, the increase in tensile strength on average for all three annealing times 1.24 N/mm^2^, at a slightly higher temperature of about 90 °C or 100 °C.

The regression models also indicated the same correlation between the annealing temperature and layer height. The coefficient next to LH × TA, which indicates the mutual dependence of these two parameters, was 0.720375677, which is much higher than any other coefficient that illustrates the dependency of any two parameters. This means that with PETGCF it is very important to consider which combination of LH and TA will be used, as on different LH values, annealing temperatures can yield very different results on tensile strength.

The base tensile strength of each material (on each layer height) was also determined with no annealing performed on the specimens. The results showed that for PLA, annealing at 100 °C for 90 min after printing with a 0.2 mm layer height provided the greatest improvement, resulting in a 6.28 % increase in tensile strength after annealing compared to the base specimen (from base of 30.07 MPa to 31.96 MPa). For PETG, annealing at 70 °C for 90 min after printing with a 0.3 mm layer height provided the greatest improvement, resulting in an 8.08 % increase in tensile strength, compared to the base specimen (from base of 28.2 MPa to 31.04 MPa). For PETGCF, annealing at 60 °C for 90 min after printing with a 0.1 mm layer height provided the greatest improvement, resulting in a 14.89% increase in tensile strength, compared to the base specimen (from base of 29.15 MPa to 33.49 MPa). All the greatest improvements for each material at each layer height are presented in Table 6.

Regarding the modulus of elasticity, it is difficult to draw definite conclusions about regularities. However, one overall trend that is visible is that PETGCF has a higher modulus of elasticity compared to PLA and PETG, regardless of annealing process, as can be seen in Figure 7. This can also be concluded for PLA material, compared to PETG, as PLA has higher modulus of elasticity, regardless of annealing process.

Similarly, as it was presented for tensile strength, one may notice the improvement of the modulus of elasticity for each material on each layer height and compare it to the base modulus of elasticity. The results indicate that for the PLA, the best improvement was achieved by annealing at 100 °C for 90 min after printing with a 0.2 mm layer height, resulting in a 12.73 % increase in modulus of elasticity compared to the base specimen (from base of 1392.99 MPa to 1570.38 MPa). For PETG, the best improvement was obtained by annealing at 90 °C for 30 min after printing with a 0.1 mm layer height (from base of 1083.83 MPa to 1140.8 MPa) and at 70 °C for 30 min after printing with a 0.2 mm layer height (from base of 1039.48 MPa to 1094.99 MPa), both resulting in an about 5% increase in modulus of elasticity compared to the base specimens.

For PETGCF, the best improvement was achieved by annealing at 100 °C for 90 min after printing with a 0.2 mm layer height, resulting in a 21% increase in tensile strength compared to the base (from base of 1710.96 MPa to 2072.48 MPa). Table 7 presents all the best improvements of modulus of elasticity for each material on each layer height.

The study also revealed interesting findings regarding the dimensional change of the specimens, as presented Table 8. Table 8 provides information on the average change (average change for all the specimens of each material, but without base specimen (the one without annealing performed on it)), standard deviation of change (standard deviation of change of the specimens used for calculating average change), and maximum change (the specimen that had the biggest dimensional change) observed in the dimensions of thickness, width, and length for each material, along with the corresponding parameter combinations of layer height, annealing time, and annealing temperature. PETGCF stands out as the material with the smallest changes in measurements among the three materials. The average change, maximum change, and even the standard deviation of changes for PETGCF are relatively low compared to PLA and PETG. This suggests that PETGCF exhibits greater dimensional stability during annealing, making it a promising material for applications where precise dimensions are crucial. PETG, on the other hand, appears to be the most unstable material, especially when considering the length dimension. Although the average changes for PLA and PETG are similar, the standard deviation of change in length for PETG is significantly higher at 4.99 mm. This indicates that PETG is more sensitive to variations in annealing temperatures and times, leading to larger and more unpredictable changes in length. This is further evident when examining the parameter combination of 0.1/90/90, which resulted in the maximum length change of 22 mm compared to the base specimen without annealing. The experiment was repeated multiple times using this combination, and similar changes in length were consistently observed, indicating reproducibility. However, the underlying reasons for this reproducibility are not fully understood. When observing the combination of 0.1mm/60min/90 °C, a change of approximately 10% (16.7 mm) was observed. This indicates that when annealing at 90 °C with a layer height of 0.1 mm, the PETG specimens are almost unusable due to dimensional changes. Regarding the parameter combinations for maximum changes, there are a few noteworthy findings. For PLA, the combination of LH/TTA/TA: 0.1/60/100 resulted in the largest change in width at 0.7 mm. For PETG, the combination of LH/TTA/TA: 0.1/90/90 led to the maximum width change of 1.25 mm. Finally, for PETGCF, the parameter combination of LH/TTA/TA: 0.3/30/100 yielded the highest width change at 0.32 mm. These parameter combinations highlight specific conditions that can induce significant changes in the width dimension for each material, and should be avoided, if dimensional accuracy is of any concern. Regarding minimal change of dimensions, some specimens did not change at all after annealing, at least for some dimensions. For PLA, all specimens exhibited some change in length and width, but specimens with combinations of 0.3 mm/90 min/90 °C and 0.3 mm/60 min/100 °C (for LH/TTA/TA) did not change in terms of thickness. For PETG specimen that did not exhibit change in length was the one with the combination of 0.3 mm/60 min/70 °C, and the one that did not exhibit change in width and in thickness was 0.3 mm/30 min/90 °C. For PETGCF, the one that did not exhibit change in length combinations was 0.3 mm/90 min/80 °C and 0.3 mm/90 min/70 °C. Additionally, 0.3 mm/90 min/60 °C and 0.3 mm/60 min/70 °C did not change at all, for width 0.3 mm/60 min/90 °C, 0.3 mm/60 min/70 °C, and 0.3 mm/30 min/90 °C did not change, and for thickness 0.3 mm/90 min/60 °C, 0.3 mm/60 min/60 °C, and 0.2 mm/90 min/80 °C did not change.

In the context of 3D printing and annealing of 3D printed parts, a trade-off between dimensional accuracy and tensile strength becomes evident. Our findings suggest that for achieving optimal dimensional accuracy, caution should be exercised when utilizing small layer heights (0.1 mm) and large annealing temperatures (90 °C, 100 °C). These factors have been observed to produce the most substantial changes in dimensions compared to the base specimens. Small layer heights, while offering finer resolution and smoother surfaces, can increase the susceptibility of the printed parts to dimensional variations during the annealing process. Similarly, larger annealing temperatures contribute to significant thermal expansion and contraction, further exacerbating dimensional changes. However, it is important to note that when considering tensile strength, smaller layer heights and larger annealing temperatures are generally more advantageous. The enhanced interlayer adhesion and improved molecular bonding resulting from smaller layer heights and higher annealing temperatures contribute to superior mechanical properties, including increased tensile strength. Thus, a careful balance must be struck between dimensional accuracy and tensile strength by considering the specific requirements of the application, ensuring that the chosen parameters for 3D printing and annealing are optimized accordingly.

## 4. Conclusions

In this study, the effect of layer thickness, different annealing temperatures, and times of annealing on the tensile strength and dimensional change of three different materials used in FDM printing (PLA, PETG, and PETGCF) was investigated under identical testing and annealing conditions. The regression models developed in this study offer a valuable tool for assessing the effect on the tensile strength of 3D-printed parts made from different materials based on input parameters. Specifically, the models consider the impact of layer thickness, annealing temperature, and annealing time on the resulting tensile strength. The main conclusions are summarized as follows:Layer height has the greatest impact on the tensile strength of 3D printed parts made from PLA, PETG, and PETGCF materials. Increasing layer height results in a decrease in tensile strength and vice versa. Average tensile strength of all specimens that were printed with layer height of 0.1 mm, 0.2 mm, and 0.3 mm for PLA is 32 MPa, 30.19 MPa, and 28.75 MPa; for PETG is 33.52 MPa, 30.37 MPa, and 29.45 MPA; and for PETGCF is 28.46 MPa, 24.396 MPa, and 22.52 MPa, which clearly displays the trend of tensile strength being higher for smaller layer heights.Annealing time has a greater impact on the tensile strength of PETGCF materials than PLA and PETG materials. Increasing annealing time results in improved tensile strength for PETGCF materials, but not so much for PLA and PETG. This trend is clear when looking at the coefficient next to TTA in regression models, where for PETGCF it is 0.506168091, and for PLA and PETG 0.186613358 and −0.13101469. This is also clear when looking at the average tensile strengths for these materials at different annealing times, where for 30 min, 60 min, and 90 min of annealing time, PLA has average tensile strength of 29.96 MPa, 30.56 MPa, and 30.42 MPa, which are very similar to each other, and PETG has 31.3 MPa, 31.04 MPa, and 30.98 MPa, which are also very similar. For PETGCF, the difference is a little bit more visible, as for 30 min, 60 min, and 90 min, average tensile strength values are 24.58 MPa, 25.25 MPa, and 25.55 MPa.Annealing temperature has an effect on the tensile strength of different materials. However, the specific range of temperatures that leads to an increase in tensile strength varies depending on the material. While PLA and PETG showed an increase in tensile strength at specific temperature ranges (PLA at 80 °C and 90 °C, and PETG at 60 °C and 70 °C); however, the effect of annealing temperature on PETGCF was not as clear. Therefore, it is crucial to carefully adjust the annealing temperature for each material to achieve the desired improvement in tensile strength. For PLA, average tensile strength for all combinations at 80 °C being 30.54 N/mm^2^, and for 90 °C being 30.57 N/mm^2^, compared to lower temperatures where for 60 °C and 70 °C, average tensile strengths were 29.49 N/mm^2^ and 29.33 N/mm^2^, respectively. PETG at temperatures of 60 °C or 70 °C gives the best results, as average tensile strength for all combinations was 31.41 N/mm^2^ and 32.1 N/mm^2^, respectively, compared to higher temperatures of 90 °C and 100 °C, where average tensile strengths were 30.73 N/mm^2^ and 30.29 N/mm^2^. PETGCF is highly dependent on layer height. For LH of 0.1 mm, the best average tensile strength (of 32.31 N/mm^2^) is at temperatures of 60 °C, compared to other temperatures, that do not go over 29 N/mm^2^. For LH of 0.2 mm, temperature of 70 °C shows the best results with average tensile strength of 25.93 N/mm^2^, with all others being around or below 24 N/mm^2^. Additionally, for LH of 0.3 mm, temperature of 100 °C has average tensile strength of 23.39 N/mm^2^, with all others being around 22 N/mm^2^. So, for PETGCF it is important to use an appropriate combination of layer height and annealing temperature, as annealing temperature causes the material different effects in different layer heights.The dimensional change of 3D printed parts during annealing varies depending on the material, layer height, and annealing parameters. PETG material showed more change than PLA and PETGCF (maximal change for PLA and PETGCF, in length was 5.8 mm and 1 mm, respectively, with PETG having maximal change of 22 mm), especially at high annealing temperatures (90 °C, 100 °C) and times (90 min) and lower layer heights (0.1 mm layer height). PETGCF demonstrated excellent resistance to dimensional change during the annealing process. As average change in width was 0.07 mm, in thickness 0.02 mm and in length 0.28 mm, with great results for standard deviations (0.07 mm, 0.02 mm, 0.27 mm) which indicates the stability in the dimensional change across all combinations. This could be to the carbon fiber that is imbedded in PETG material, which gives it higher temperature resistance.For modulus of elasticity, one overall trend that was noticed is that PETGCF has the highest modulus of elasticity across all combinations, then PLA, and lastly PETG. The study found that annealing PLA at 100 °C for 90 min after printing with a 0.2 mm layer height led to a significant improvement in modulus of elasticity, with a 12.73% increase compared to the base specimen (from 1392.99 MPa to 1570.38 MPa). Similarly, for PETG, annealing at 90 °C for 30 min after printing with a 0.1 mm layer height, and at 70 °C for 30 min after printing with a 0.2 mm layer height, resulted in approximately a 5% increase in modulus of elasticity (from 1083.83 MPa to 1140.8 MPa and from 1039.48 MPa to 1094.99 MPa, respectively). Lastly, for PETGCF, annealing at 100 °C for 90 min after printing with a 0.2 mm layer height led to a significant 21% increase in tensile strength compared to the base specimen (from 1710.96 MPa to 2072.48 MPa).The findings suggest that when annealing 3D printed specimens, the layer height (LH) has the most significant influence on the resulting tensile strength. Changes in the annealing time (TTA) and annealing temperature (TA) have negligible effects on tensile strength compared to LH. Regarding modulus of elasticity, PETGCF consistently exhibits the best performance across all combinations of LH, TTA, and TA, followed by PLA and PETG. Furthermore, incorporating dimensional changes after annealing reveals a trade-off between dimensional accuracy and tensile strength. Optimal dimensional accuracy is achieved by using higher LH (e.g., 0.3 mm) and smaller TA (e.g., 30 min), which contrasts with the preference for smaller LH (e.g., 0.1 mm) and higher TA (90 min) to enhance tensile strength. Smaller layer heights, while providing finer resolution and smoother surfaces, can increase the susceptibility of printed parts to dimensional variations during the annealing process. Additionally, larger annealing temperatures contribute to significant thermal expansion and contraction, further exacerbating dimensional changes. The combination of smaller layer heights and higher annealing temperatures leads to enhanced interlayer adhesion and improved molecular bonding, resulting in superior mechanical properties, including increased tensile strength.Overall recommendations of the study: because TA and TTA do not contribute much to tensile strength, it can be suggested that if in need of tensile strength of FDM 3D printed parts, one should just use smaller LH values, as annealing process does not give significantly better results. However, if one uses annealing process, PLA combination of 0.1 mm/60 min/90 °C (LH/TTA/TA) should be used as it gave the best results of tensile strength (33.37 MPa), for PETG and PETGCF one should use 0.1 mm/90 min/60 °C as this combination gave the best results of tensile strength (35.6 MPa and 33.49 MPa, respectively) for both materials. Regarding the materials, overall PETGCF did not perform well in terms of tensile strength, so it is better to use PLA or PETG material. Dimensional change of PETGCF, which was much lower after annealing than the ones of PLA and PETG can be neglected if one follows previous recommendation (not using annealing), because there will be no annealing and non-change in dimensions will be present. However, if elasticity is one of the required characteristics of the material, PETGCF is a better option than PLA or PETG.Typically, achieving optimal printing parameters for a given material ensures that the final part does not necessitate additional post-processing treatments, such as heat treatment, to enhance mechanical properties. However, selecting optimal printing parameters can be challenging, particularly for novel materials in the field. PLA and PETG are extensively utilized materials in FDM, with well-defined and provided printing parameters by manufacturing companies. It is plausible that adhering to these recommended printing parameters could be a primary factor contributing to minimal property enhancements or even diminished properties when conducting annealing processes with unsuitable heat treatment conditions.

This paper provides a comprehensive analysis of the effects of layer height and annealing process on mechanical and dimensional properties of 3D printed parts. However, the study only assessed these factors individually, and there is still needed an approach that considers multiple criteria simultaneously. As such, future work will involve using multi-criteria decision-making techniques to evaluate the optimal combination of layer height and annealing process based on criteria such as maximum tensile strength, maximum modulus of elasticity and minimum dimensional change (and more conditions can be added). In addition, it can also be beneficial to utilize the regression models developed in this study to further optimize the 3D printing process and determine the absolute best set of input parameters for achieving optimal mechanical and dimensional properties.

## Figures and Tables

**Figure 1 materials-16-04574-f001:**
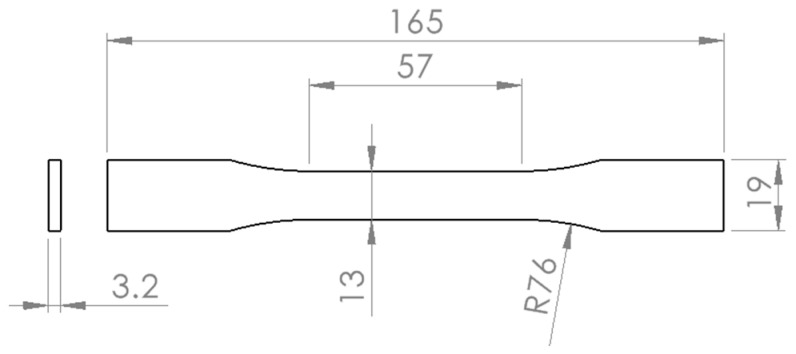
Dimensions of specimen compliant with ASTM D638-14 standard.

**Figure 2 materials-16-04574-f002:**
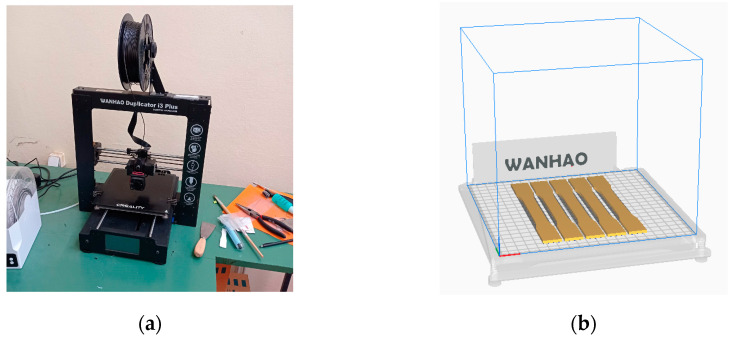
The 3D printer used in the experiments: (**a**) Wanhao Duplicator i3 Plus 3D printer and (**b**) batches of five specimens in Ultimaker Cura 5.2.2. slicing software.

**Figure 3 materials-16-04574-f003:**
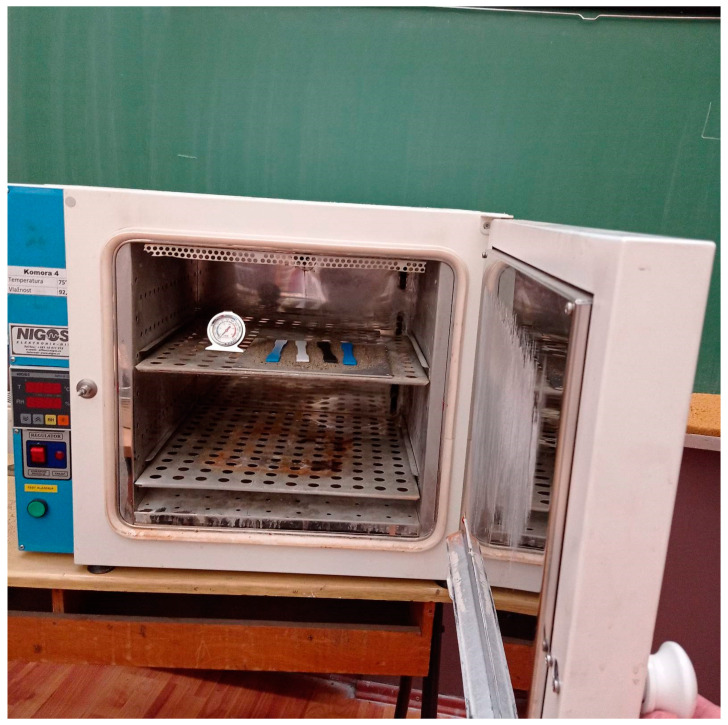
Industrial oven and thermostat used for annealing of the printed specimens.

**Figure 4 materials-16-04574-f004:**
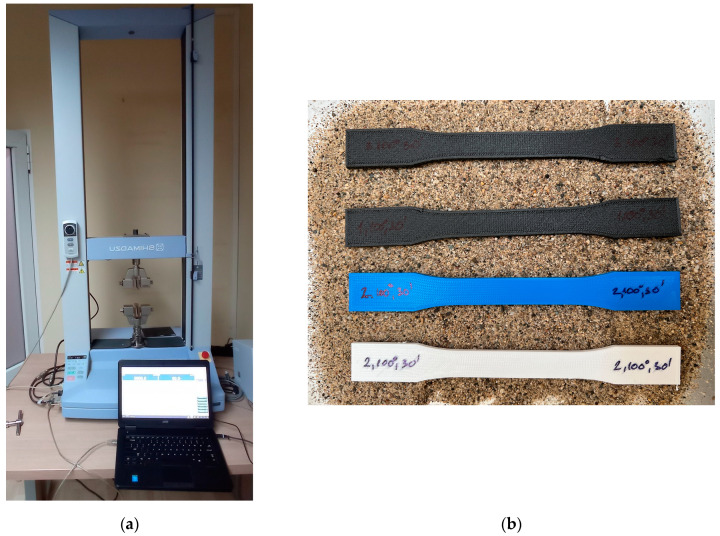
Experimental setup: (**a**) Shimadzu Table-top AGS-X 10 kN universal testing machine and (**b**) some of the specimens to be tested.

**Figure 5 materials-16-04574-f005:**
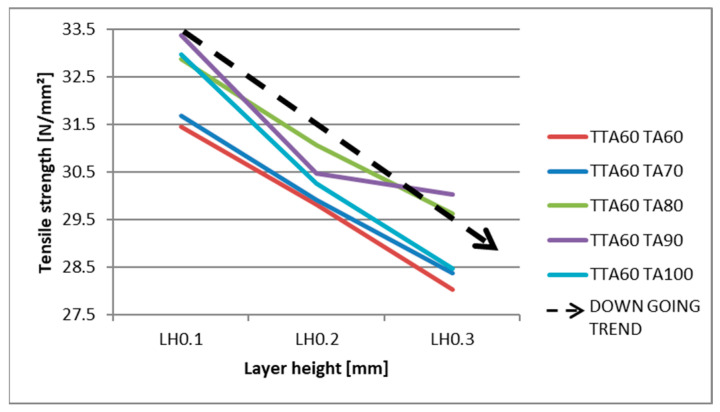
Effect of the layer height on the tensile strength (for PLA material).

**Figure 6 materials-16-04574-f006:**
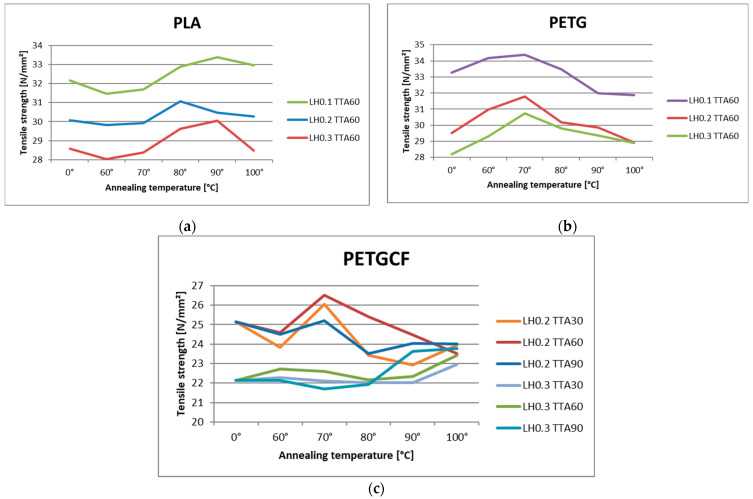
Influence of annealing temperature on tensile strength: (**a**) for PLA material; (**b**) for PETG material; and (**c**) for PETGCF material.

**Figure 7 materials-16-04574-f007:**
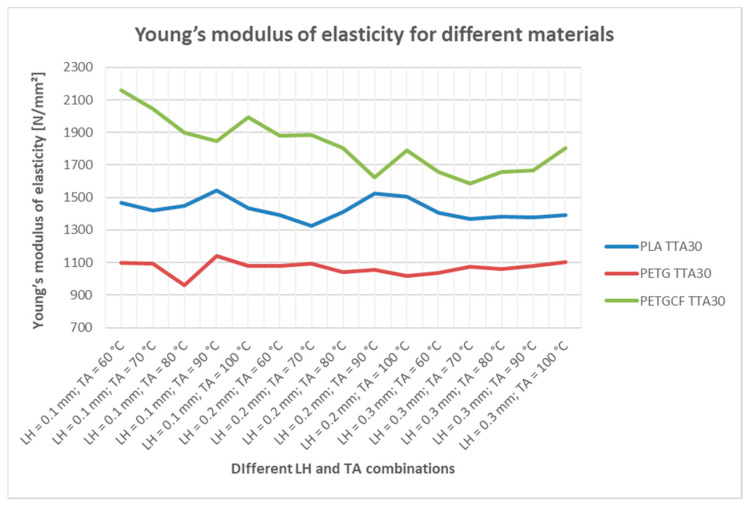
Modulus of elasticity for different materials.

**Table 1 materials-16-04574-t001:** Overview of scientific papers that studied annealing process and its effects on mechanical properties of 3D-printed parts.

#	Summary	Results
[45]	Thermal annealing was applied to printed PLA parts at different temperatures and for different durations	Maximum flexural stress increased by 17% and 11% for samples annealed at 85 °C for 70 min and 95 °C for 15 min, respectively, compared to non-annealed samples. The flexural strain did not show any significant change.
[46]	Investigated the impact of thermal annealing on the interlayer tensile strength of FDM-printed PLA specimens. Two annealing temperatures were applied (90 °C and 120 °C) for PLA, PLACF, PETG, and PETGCF, and three annealing times (30, 240, and 480 min)	Annealing at 120 °C for 30, 240, and 480 min did not significantly affect interlayer tensile strength. Annealing at 90 °C for 30, 240, and 480 min led to a slight increase in interlayer tensile strength. However, increasing the annealing time at 90 °C did not significantly improve interlayer tensile strength.
[47]	Investigated the effect of thermal annealing on the tensile strength of FDM-printed PLA, ABS, Cu-PLA, and Al-ASA specimens. The annealing was performed at temperatures of 70 °C, 80 °C, and 90 °C for 60 min for PLA and Cu-PLA, while for ABS and Al-ASA, higher temperatures were used due to their glass transition temperatures.	For PLA specimens the increase in tensile strength was more significant when the annealing temperature increased from 70 °C to 80 °C, while further increasing the annealing temperature to 90 °C had no additional effect on the tensile strength of the specimens. The amorphous materials, ABS, and Al-ASA did not show a significant improvement in tensile strength.
[48]	Tested three variations of polylactic acid material and annealed them at temperatures from 60 to 160 °C for 30 min.	PLA-HD showed the highest percentage increase (9.62% at 80 °C) and PLA-PLUS showed the lowest increase (2.87% at 80 °C). PLA material showed the highest deformation values and dependence on the annealing temperature.
[49]	Conducted a study on the impact of thermal annealing on the tensile strength of FDM-printed PLA specimens. The study involved annealing the specimens at 80 °C and 100 °C for durations of 30, 60, and 120 min.	Maximum increase in TS was achieved when annealing the specimens at 80 °C for 60 min. Changing the holding time from 30 to 60 min or from 60 to 120 min had no significant effect on the TS of the specimens when annealed at 80 °C. Varying the annealing temperature and time did not significantly affect the TS of the specimens within the temperature and holding time ranges tested in the study.
[50]	Investigated the effects of thermal annealing on the mechanical properties of PLA. Three different temperatures for annealing were selected (90 °C, 100 °C, and 120 °C) and three different time periods (60 min, 120 min, and 240 min)	Results showed that the mechanical properties of the PLA, particularly its tensile properties, could be significantly enhanced by heat treatment at around 100 °C for 4 h. Specifically, they found that the tensile properties of the PLA could be improved by up to 80% through this annealing process.
[51]	Investigated the impact of printing orientation, layer height, and thermal annealing on the tensile properties and anisotropy of PLA FDM parts. Tested layer heights included 0.24 mm, 0.16 mm, and 0.12 mm. Thermal annealing was performed at 60 °C for 1 h.	Reducing the layer height from 0.24 mm to 0.16 mm improved PLA FDM specimens’ tensile modulus by 12% and tensile strength by 67% in the ZXY orientation. However, print time increased by 91%. Similarly, using a 0.12 mm layer height resulted in a 71% higher tensile strength and a 16% higher tensile modulus compared to 0.24 mm layer height, but print time increased by 154%. Annealing increased PLA parts’ strength by 24%, but caused deformation, limiting its industrial applicability.
[52]	Studied the interlaminar toughness of polymers with annealing.	The interlaminar toughness of polymers can be increased by annealing, resulting in better performance than injection molding samples
[53]	The impact of annealing on PETG and CFPETG composites with different infill densities (25%, 50%, 75%, and 100%) was analyzed. Annealing was performed at a temperature of 5 degrees Celsius above the glass transition temperature of the materials for a duration of 60 min.	Results showed that annealed specimens with 100% infill density exhibited the highest mechanical properties. Annealed CFPETG at 100% infill density showed a 10–11% increase in mechanical properties compared to as-printed CFPETG, while annealed PETG at 100% infill density demonstrated a 6–8% increase compared to as-printed PETG. CFPETG samples outperformed PETG samples with improvements of 21% in hardness, 25% in tensile strength, 23% in impact strength, and 18% in bending strength.
[54]	Studied the effects of annealing on 3D printed parts with different layer heights (0.10, 0.15, 0.20 mm) and infill percentage (50%, 75%, 100%).	PLA samples showed enhanced mechanical properties after a 3 h heat treatment at 75 °C. Using layer thicknesses of 0.10 mm, 0.15 mm, or 0.20 mm and infill percentages of 50%, 75%, or 100% led to improved mechanical properties. Annealed samples exhibited an average increase of approximately 30% in tensile strength compared to non-annealed samples, regardless of the infill percentage.
[55]	Investigated the impact of infill patterns and annealing on the mechanical properties of PETG and CFPETG parts produced using FDM. They tested four different infill patterns (grid, honeycomb, rectilinear, and cubic) and annealed the samples at 100 °C for 60 min.	The study found that annealing the PETG and CFPETG parts produced with the grid infill pattern resulted in the highest improvement in stiffness, tensile strength, impact strength, and flexural strength, with increases of 29%, 27%, 18%, and 9%, respectively. Combining the printed grid infill pattern with annealing the PETG and CFPETG parts resulted in a 17% increase in tensile strength.
[56]	The effect of annealing treatment was investigated on composites of PLA, ABS, and PETG reinforced with 13.2%, 14.4%, and 17.2% CF by weight, respectively, along with non-reinforced PLA, ABS, and PETG, with three different infill patterns. The authors’ utilized three different annealing temperatures (65 °C, 110 °C, and 85 °C) based on the different glass transition temperatures of the matrices.	The study found that carbon fiber increased Young’s Modulus and flexural modulus but had no significant effect on tensile or flexural strength in CF-PLA, CF-ABS, and CF-PETG specimens. Annealing for 60 min resulted in average increases of 1%, 6.5%, −6.2%, −0.8%, 7.84%, and 9.74% (ranging from 2.78% to 13%) in tensile strength for PLA, PLACF, PETG, PETGCF, ABS, and ABSCF, respectively. Annealing improved tensile strength, tensile stiffness, and flexural strength for PLA, CF-PLA, PETG, and CF-PETG, but reduced flexural stiffness. Annealing had mixed effects on ABS and CF-ABS and was beneficial for specific infill patterns.
[57,58]	Used three temperatures (90 °C, 110 °C, and 130 °C) and three annealing times (30 min, 240 min, and 480 min) to study the effects of thermal annealing on the bending properties of PETG and PETG reinforced with carbon and aramid fibers.	The researchers found that increasing both the temperature and exposure time resulted in a significant increase in flexural strength and modulus. However, the study also found that higher temperatures and longer exposure times resulted in greater geometric distortions, indicating that a temperature of 90 °C and an exposure time of 30 min were more effective in improving the mechanical properties of the materials studied.
[59]	Conducted a study on the effects of thermal annealing on the mechanical properties of a PLA-CF (carbon fiber) composite. They carried out annealing at 4 different temperatures (65 °C, 95 °C, 125 °C and155 °C) for durations of 30, 60, 120, and 240 min.	The results showed that annealing at 95 °C for 120 min led to a 14% increase in the tensile strength of the material.
[60]	studied how annealing affects the mechanical properties of blends made of (PLA) and (PCL).	They observed that the bending strength of the blends increased after annealing. Additionally, both the bending strength and modulus of the PLA/PCL blends showed improvement following annealing.
[61]	Investigated the effects of annealing process on tensile strength of 3D-printed parts, when the annealing temperature is near glass transition temperature.	Thermal annealing above the glass transition temperature enhances the ultimate tensile strength of FDM-printed polylactic acid parts. Holding time variations from 45 to 75 min have negligible effects on tensile strength. However, for optimal crystallization at temperatures between 65 °C and 85 °C, longer holding times are required.

**Table 2 materials-16-04574-t002:** All combinations of specimens (LH—layer height; TTA—annealing time; TA—annealing temperature). These combinations were performed for all three materials (PLA, PETG, PETGCF).

LH (mm)	TTA (min)	TA (°C)	LH (mm)	TTA (min)	TA (°C)	LH (mm)	TTA (min)	TA (°C)
0.1	0	0	0.2	0	0	0.3	0	0
0.1	30	60	0.2	30	60	0.3	30	60
0.1	30	70	0.2	30	70	0.3	30	70
0.1	30	80	0.2	30	80	0.3	30	80
0.1	30	90	0.2	30	90	0.3	30	90
0.1	30	100	0.2	30	100	0.3	30	100
0.1	60	60	0.2	60	60	0.3	60	60
0.1	60	70	0.2	60	70	0.3	60	70
0.1	60	80	0.2	60	80	0.3	60	80
0.1	60	90	0.2	60	90	0.3	60	90
0.1	60	100	0.2	60	100	0.3	60	100
0.1	90	60	0.2	90	60	0.3	90	60
0.1	90	70	0.2	90	70	0.3	90	70
0.1	90	80	0.2	90	80	0.3	90	80
0.1	90	90	0.2	90	90	0.3	90	90
0.1	90	100	0.2	90	100	0.3	90	100

**Table 3 materials-16-04574-t003:** Specifications of the Wanhao Duplicator I3 Plus 3d printer.

Printing technology	FDM (FFF)
Build volume	200 × 200 × 180 mm
Layer resolution	0.1–0.4 mm
Extruder number	1
Nozzle diameter	0.4 mm
Printing speed	10–100 mm/s
Max extruder temperature	260 °C
Max print bed temperature	100 °C

**Table 4 materials-16-04574-t004:** Printing parameters for different materials used in the experiment.

Printing Parameter	Value for PLA	Value for PETG	Value for PETGCF
Extruder temperature (°C)	210	235	235
Print bed temperature (°C)	60	73	73
Printing speed (mm/s)	50	50	50
Wall thickness (mm)	0.6	0.6	0.6
Top thickness (mm)	0.6	0.6	0.6
Bottom thickness (mm)	0.6	0.6	0.6
Infill percentage (%)	75	75	75
Infill pattern	Cubic	Cubic	Cubic

**Table 5 materials-16-04574-t005:** Specifications of the Shimadzu Table-top AGS-X 10 kN universal testing machine.

Brand	Shimadzu
Model	Table-top AGS-X 10 kN
Weight	85 kg
Power	1.2 kW
Max load/capacity	10 kN
Dimensions	W653 × D520 × H1603 mm
Crosshead speed range	0.001 to 1000 mm/min
Crosshead speed accuracy	0.1%
Crosshead–table distance (tensile stroke)	1200 mm (760 mm, MWG)
Data capture rate	1000 Hz max

**Table 6 materials-16-04574-t006:** Biggest improvements in tensile strength for each material on each layer height.

Materials	Parameter Combination (LH/TTA/TA)	% of Improvement Compared to Base Tensile Strength	Improvement in MPa, Compared to Base Tensile Strength
PLA	0.1/60/90	3.79%	1.22
	0.2/90/100	6.28%	1.89
	0.3/60/90	5.14%	1.47
PETG	0.1/90/60	7%	2.33
	0.2/60/70	7.66%	2.26
	0.3/90/70	8.08%	2.84
PETGCF	0.1/90/60	14.89%	4.34
	0.2/60/70	5.49%	1.38
	0.3/90/100	7.58%	1.68

**Table 7 materials-16-04574-t007:** Maximum improvements in modulus of elasticity for each material at different layer heights.

Materials	Parameter Combination (LH/TTA/TA)	% of Improvement Compared to the Base Modulus of Elasticity	Improvement in MPa, Compared to Base Modulus of Elasticity
PLA	0.1/90/100	4.89%	72.68
	0.2/90/100	12.73%	177.39
	0.3/60/90	3.17%	44.87
PETG	0.1/30/90	5.25%	56.97
	0.2/30/70	5.34%	55.51
	0.3/60/100	2.58%	27.89
PETGCF	0.1/90/100	11.83%	232.38
	0.2/90/100	21.13%	361.52
	0.3/90/100	10.68%	173.2

**Table 8 materials-16-04574-t008:** Average and maximum changes of 3D printed specimens, for all tested materials.

Material	Measure	Average Change (mm)	Standard Deviation of Change (mm)	Max Change (mm)	Parameter Combination for Max Change (LH/TTA/TA)
PLA	Δ*W*	0.22	0.16	0.7	0.1/60/100
	Δ*T*	0.1	0.06	0.25	0.1/30/100
	Δ*L*	3.55	1.19	5.8	0.1/90/100
PETG	Δ*W*	0.17	0.24	1.25	0.1/90/90
	Δ*T*	0.11	0.16	0.76	0.1/90/90
	Δ*L*	3.33	4.99	22	0.1/90/90
PETGCF	Δ*W*	0.07	0.07	0.32	0.3/30/100
	Δ*T*	0.02	0.02	0.08	0.3/60/100
	Δ*L*	0.28	0.27	1	0.1/90/100

## Data Availability

Not applicable.

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
