# Peer review of "An Experimental Study on the Impact of Layer Height and Annealing Parameters on the Tensile Strength and Dimensional Accuracy of FDM 3D Printed Parts"

_materials, 2023, doi:10.3390/ma16134574_

Round 1

Reviewer 1 Report

The article presents an analysis of the annealing process of elements made with 3D printing technology, it is current and scientifically significant. The influence of annealing on the strength properties of the samples was evaluated, and the elasticity modulus of the samples was determined.
The article is written in a technically correct layout, but it requires clarification by the authors of several important threads, also making some improvements to the work.
The article contains a large simplification in the area of selection of parameters for annealing the material, the author does not mention anything about it. The selection of the proper annealing temperature, heating time (speed), annealing time, cooling time has a significant impact on the preservation of the crystallinity phase of the material as a consequence of its storage modulus.
These parameters (temperature) should be determined on the basis of material - catalog data of the manufacturer. The annealing time should be determined based on the size - volume/thickness of the material. This is closely related to the enthalpy of melting of the crystalline phase, which in turn affects the research carried out by the authors and helps in drawing correct scientific conclusions.
The work should specify the range of melting point of the crystalline phase - this has a significant impact on the selection of appropriate annealing temperatures.
How did the author - on what basis did he choose the heating temperatures?
Figure 1, - it is rather a table, no units in the table, it should be normalized in accordance with the requirements of the magazine - table editing.
Figure2 - titled "geometry" - this is not a geometer, I recommend developing a drawing with all its dimensions marked in accordance with the indicated standard.
Figure 5a, I recommend the author not to place the surroundings of research devices - DOORS so exposingly. More effort can be made to make the graphic design of the work better.
The article raises measurement doubts, figure 6 should not be presented in a scientific work, it does not constitute a significant reference to the research method, moreover, the measurement of the thickness of the sample made as in photo a) is incorrect. You cannot measure a caliper with flat jaws adjacent to the measuring surface in this way, these are the basis for using the basic measuring device - calipers, presenting such photos does not prove the author's engineering skills - I recommend removing F6 completely. The measurement of the thickness of the sample should preferably be made with a micrometer with conical measuring tips.
The author analyzes the dimensional deviation after annealing, it is not clear in relation to what - the nominal size of the sample before annealing or in relation to the assumed structural dimension?
Table 7, what does "average change" "standard deviation of change" max change mean? in the context of the measuring device used?
Calipers with a measurement accuracy of 0.02mm ?? This should be standardized in terms of the measuring device used.
Please be more specific, the printer will print samples with different accuracy, depending on the printing parameters - each sample will have different deviations, therefore reference of changes in deviations after annealing must be carried out in relation to the nominal dimensions of the sample in relation to the same sample. If so, this is not a very big simplification, did the author take this into account when planning the research?
figure 5 dimensional deviations is illegible, it is not clear what the individual results refer to and why they differ so much from each other? The percentage deviation is illogical, did the author statistically recalculate the percentage in relation to the cumulative deviation (plus and minus)? is it a deviation from the nominal dimension? The readability of the drawing is also very low, the descriptions are not unified.
Please explain such large discrepancies in the readings for individual samples - 12% for length dimensions ???? is a 20mm deviation, the drawing is absolutely improved
figure 11 should not be included in a scientific work, it does not present anything in its form, deviations can be presented in a table or on a graph.

Spelling style and grammar are correct, a few editorial errors can be corrected at a later stage of article review.

Author Response

The authors wish to thank the reviewer for the valuable made observations. Responses to the reviewer were provided in the attached document along with the manuscript in which all corrections / new inputs were marked with red color in the text.

Reviewer 2 Report

The abstract's opening sentences are unsuitable for this section and should be removed and addressed in the introduction section. Because in general, heat treatment in the FDM process is to improve the quality of the layer connection, but this happens if the heat treatment conditions are chosen appropriately and optimally. For this purpose, the most critical issue is the selection of annealing conditions, such as temperature and time. How are these parameters selected in the current research?

Why is the temperature and annealing conditions chosen for both materials the same? While PLA is a semi-crystalline thermoplastic with a transition temperature of about 65 to 70 degrees Celsius, and on the other hand, PETG is a fully amorphous thermoplastic and its transition temperature is about 80 to 85 degrees Celsius.

The abstract needs to be reformed and innovation, the purpose of the research, the examined tests, achievements, quantitative results and improvements should be presented. Use the following reference to deepen the introduction to the application of the FDM process. 4D printing of PLA-TPU blends: Effect of PLA concentration, loading mode, and programming temperature on the shape memory effect. Mechanical performance and damage monitoring of CFRP thermoplastic laminates with an open hole repaired by 3D printed patches. 3D printed nanofiltration membrane technology for waste water distillation.

Errorbar should be added to all numerical results of mechanical properties. Why have microscopic images not been used to check the density of microholes and the quality of connections? Not considering this issue causes errors in analyzing quantitative results of mechanical properties. Because if the density of microholes is high or the quality of the connection is not favorable, heat treatment should be used, which is justified. To add these items to the article, use the following sources. Development of Pure Poly Vinyl Chloride (PVC) with Excellent 3D Printability and Macroand MicroStructural Properties. Shape memory performance of PETG 4D printed parts under compression in cold, warm, and hot programming.

In general, the selection of optimal printing parameters makes the final part not require additional treatment, such as heat treatment, to improve mechanical properties, and this is done for new materials in this field, which are challenging to select the optimal printing parameters. PLA and PETG are the most widely used materials for FDM, and the manufacturing company clearly defines and provides their printing parameters. This factor can be the main reason for the loss of properties by performing the annealing process and selecting inappropriate heat treatment conditions.

No comment.

Author Response

(The authors gave the same response as above.)
